# HybridFlow: Infusing Continuity into Masked Codebook for Extreme Low-Bitrate Image Compression

## ABSTRACT

This paper investigates the challenging problem of learned image compression (LIC) with extreme low bitrates. Previous LIC methods based on transmitting quantized continuous features often yield blurry and noisy reconstruction due to the severe quantization loss. While previous LIC methods based on learned codebooks that discretize visual space usually give poor-fidelity reconstruction due to the insufficient representation power of limited codewords in capturing faithful details. We propose a novel dual-stream framework, HyrbidFlow, which combines the continuous-feature-based and codebook-based streams to achieve both high perceptual quality and high fidelity under extreme low bitrates. The codebook-based stream benefits from the high-quality learned codebook priors to provide high quality and clarity in reconstructed images. The continuous feature stream targets at maintaining fidelity details. To achieve the ultra low bitrate, a masked token-based transformer is further proposed, where we only transmit a masked portion of codeword indices and recover the missing indices through token generation guided by information from the continuous feature stream. We also develop a bridging correction network to merge the two streams in pixel decoding for final image reconstruction, where the continuous stream features rectify biases of the codebook-based pixel decoder to impose reconstructed fidelity details. Experimental results demonstrate superior performance across several datasets under extremely low bitrates, compared with existing single-stream codebook-based or continuous-feature-based LIC methods.

## CCS CONCEPTS

• **Computing methodologies → Image compression**.

## KEYWORDS

HybridFlow, extreme low-bitrate image compression, masking

## 1 INTRODUCTION

The explosive amount of visual information required by increasingly sophisticated applications like communication, broadcasting, gaming, *etc.* poses great challenges to network transmission and data storage. Effective image compression at ultra-low bitrates has become highly desired but remains poorly solved.

Powered by trained neural networks, learned image compression (LIC) has been proven superior than conventional methods like

**Unpublished working draft. Not for distribution.**

VVC [6] or JPEG2000 [42]. The whole idea is to encode the input image into a latent space in the encoder, compress the latent feature to reduce transmission bits, and reconstruct the output image using decompressed latent in the decoder. Based on the type of information to transfer, LIC methods can be roughly grouped into two categories. The first category [3, 9, 13, 16, 22–24, 26, 28, 32, 37, 44, 47] has been broadly studied, featuring transmitting continuous compressed feature maps. The original latent feature goes through classic quantization and entropy coding to obtain a compact bitstream with continuous values, and the decoder recovers a degraded latent feature for reconstruction. When the bitrate is extremely low, the recovered latent feature has poor quality due to severe quantization, resulting in low-quality reconstruction that is overly smooth and lacks of expressive details.

The second category [20, 21, 35] has recently merged attributed to the increasing popularity of using learned general image priors by a quantized-vector-based codebook for image restoration tasks, featuring transmitting integer indices. A learned visual codebook is pretrained to discretize the distribution of the image latent into a finite discrete set space. By sharing the codebook among the encoder and decoder, the encoder maps the latent feature into codeword indices, and the decoder recovers an approximate latent feature for reconstruction by retrieving codeword features using the integer indices. High-quality codebooks learned from high-quality images usually ensure high-perceptual-quality reconstruction with clear and rich details [7, 38, 40, 48]. However, the output image may be unfaithful to the original input, *e.g.*, small content changes are dissolved by discretized codebook. Large codebook capturing detailed visual aspects or multiple codebooks each focusing on class-specific representations [33] can alleviate this issue, but with a sacrifice of increased bitrates. Hence, when bitrate is extremely low, the limited codebook size results in poor-fidelity reconstruction.

In this paper, we propose a hybrid framework that benefits from the dual-stream complementarity of the above two categories, enabling extreme low-bitrate transmission and high-quality reconstruction at the same time. Two parallel flows are generated from the input image: a discrete index map based on a high-quality codebook that utilizes learned general image priors to obtain high-perceptual reconstruction quality, and an extreme low-bitrate continuous feature stream providing fidelity details. The two flows are combined through an effective bridging mechanism for masked token generation and corrective pixel decoding. Our contributions are:

- We introduce a novel dual-stream framework for image compression, HybridFlow, which achieves clear and faithful high-quality image reconstruction even at extremely low bitrates (<0.05 bpp), surpassing previous approaches.
- A "masked-prediction" strategy is further introduced to the codebook-based discrete flow. Motivated from the masked token-based transformer architecture of MAGE [31], by guided-generation from just a portion of the index map, we not only

reduce the transmitted indices, but also achieve a controllable trade-off between reconstruction quality and bitrate.

- We propose a bridging mechanism to merge the two information streams. The continuous features are fed into the cross-attention module of the token decoder to guide the predictive generation of the codebook-based features. At the same time, the continuous features rectify biases of the pixel decoding process using codebook-based features through a correction network alongside the pixel decoder.

We conduct experiments to evaluate the effectiveness of our approach both qualitatively and quantitatively over several benchmark datasets. Qualitative results show that our "HybridFlow" framework can preserve the high quality and clarity of codebook-based reconstruction, while effectively correcting pixel-level distortion through infusing continuous image feature. Quantitative performance achieves an average improvement of about 3.5dB in PSNR compared to pure codebook-based methods with the same or even better LPIPS scores, and a significant improvement of LPIPS scores (55.7%) compared to pure continuous-feature-based methods. Overall, our "HybridFlow" provides a balance between credibility and clarity, between trustworthiness and perceptual quality.

## 2 RELATED WORK

### 2.1 LIC based on Continuous Features

Image compression using neural network models have gained widespread attention. LIC has three key processing steps: learning a concise latent image representation; effectively quantizing this representation to reduce information transmission rates; and efficiently reconstructing high-quality images from the quantized information.

The majority of works are based on the hyperprior framework [3, 10, 16, 23, 36]. The image latent feature is compressed by the conventional quantization and entropy encoding process into a bit-efficient data stream, and the decoder recovers the degraded latent feature by conventional entropy decoding and dequantization. The transmitted data are complex floating-point numbers, and we categorize these methods as LIC based on continuous features.

Quantization is necessary to reduce transmission demands but introduces information loss in the recovered latent feature. Many research works have been focused on reducing such information loss by improving the entropy model of quantization/dequantization. These approaches usually work well for moderate to high bitrates, where quality latent feature can be recovered. However, for extremely low bitrates, heavy quantization leads to significant degradation in recovered latent feature, resulting in severe reconstruction artifacts like blurs, blocky effects, etc.

### 2.2 LIC based on Codebooks

Learned generative image priors, a.k.a., visual codebooks, have achieved impressive performance over a variety of image restoration tasks [8, 11, 17, 18, 33, 43, 48]. The visual codewords span a quantized latent feature space into which each image can be embedded as a quantized feature by mapping to the nearest codewords. This idea naturally aligns with the compression task, and has been used for LIC recently [20, 21, 35]. The encoder computes the embedded quantized feature, which is represented by an integer indices map of mapped codewords, and the decoder retrieves the quantized

feature from the indices map using the same codebook. The integer indices are extremely efficient and robust to transfer, and we categorize these methods as LIC based on codebooks.

The richness of the codebook, e.g., the number of multiple codebooks capturing different semantic visual subspaces like AdaCode [33], the size of codebooks, and the information required to combine multiple codebooks [21], determines the tradeoff between bitrate and reconstruction quality. Due to the difficulty of learning a universally abundant visual codebook to capture the complicated general image content, such methods usually show advantages with low to moderate bitrates, unless when applied to specific content like human faces [20, 48]. For general high-bitrate cases, the finite number of codewords can be insufficient to recover the rich details of the general latent feature, in comparison to the hyperprior framework with conventional quantization. With low to moderate bitrates, the high-quality codebook can give high-quality latent feature for high-quality reconstruction, despite the inputs' quality. However, when the bitrate is extremely low, the small codebook causes too much collapse in the visual space. As a result, different images can be treated as variants of each other and mapped to the same sets of codewords, giving similar generic reconstruction. In such cases, although the output has high perceptual quality, it can be pixel-level (even semantic-level) unfaithful to the original input.

### 2.3 Masked Image Modeling

Masked Image Modeling aims at utilizing self-supervised learning to improve the efficiency and robustness of the learned image representation. The key idea is to remove a portion of the input before passing it to the model and training the model to reconstruct the missing content. Early researches like MAE [15], BEiT [4], ADIOS[41] , SiamMAE [14] and SimMIM [46] directly work on the pixel level. This usually leads to blurred reconstruction (e.g, as shown in MAE sample figures), since pixels are considered low-level representations rather than high-level semantics and it is innately challenging to learn high-level representations by random masking. To inject more semantics into target tokens, various works based on discrete image representations have been purposed recently. For example, when an image is encoded into a token sequence in a finite discrete space, a tokenizer similar to language models, is applied to the masking modeling. Masked Generative Encoder (MAGE) [31] (an extension of MAE in the discrete latent domain) and MaskGIT [7] demonstrate remarkably strong ability to learn image representations. Such methods address previous issues in image domain modeling, such as blurriness and excessive smoothing.

Nevertheless, we have noticed a phenomenon where these models, when provided with different partial information from the same original image, tend to generate visually dissimilar images. Although these generated images exhibit semantic similarity and clarity based on the learned distribution, they diverge significantly from the visual characteristics of the original image. This discrepancy contradicts the objective of maintaining fidelity to the original image in the context of image compression tasks.

## 3 METHOD

Figure 1 gives the workflow of our dual-stream HybridFlow for high-quality reconstruction with ultra low bitrates (< 0.05bpp).

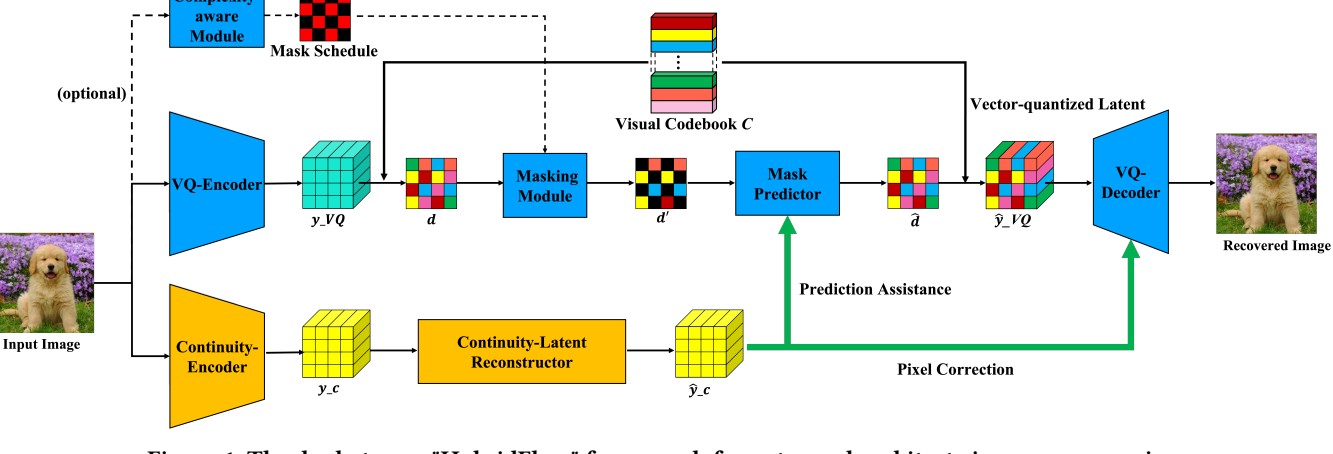

**Figure 1: The dual-stream "HybridFlow" framework for extreme low-bitrate image compression.**

## 3.1 HybridFlow Image Compression

**Codebook-based Representation.** Given an input image $x \in \mathbb{R}^{3 \times H \times W}$, the first data stream is generated as a discrete indices map $d \in \mathbb{R}^{\frac{H}{n} \times \frac{W}{n}}$ by using a learned visual codebook $\mathbf{C} \in \mathbb{R}^{c \times n_z}$. Specifically, $x$ is encoded by a VQ-Encoder $\mathbf{E_{VQ}}$ into a latent representation $y_{VQ} \in \mathbb{R}^{c \times \frac{H}{n} \times \frac{W}{n}}$, which is further mapped into indices map $d$. Each entry vector $y_{ij} \in y_{VQ}$ ($i = 1, \dots, \frac{H}{n}$, $j = 1, \dots, \frac{W}{n}$) is mapped to the closest codeword $c_{ij} \in \mathbf{C}$ with a codeword index $d_{ij} \in [0, n_z)$. In practice a single codebook with a relatively small size is used to obtain an ultra low bitrate ($n_z = 1024$ with about 0.02 bpp for discrete data stream in our experiments).

**Masking Module.** To further reduce transmission bits in the discrete stream, instead of directly transmitting the indices map $d$ as previous efforts [21, 35], we opt for a selective approach to transfer only a portion of $d$. Utilizing a mask module equipped with predefined, structured, and compression-friendly masking schedules, such as 1_4 masking and 1_9 masking as illustrated in Figure 3, we transmit a masked $d'$ instead of $d$. The chosen schedules determine the proportion of the remaining information, providing more efficient compression rates.

**Continuous-domain Representation.** A second data stream is generated from the input $x \in \mathbb{R}^{3 \times H \times W}$ as a continuous latent feature $y_c \in \mathbb{R}^{f \times \frac{H}{m} \times \frac{W}{m}}$, by using a continuous-feature-based LIC method. In this paper, we use the MLIC [23] method which gives the state-of-the-art low-bitrate compression performance. In general, the lowest bitrate previous methods can provide with a reasonable reconstruction is around 0.1 bpp. To obtain an ultra low bitrate, the original image is first downsampled by 4× before fed into the MLIC pipeline. This gives about 0.025 bpp for the continuous data stream in our experiments.

**Mask Predictor.** On the decoder side, a token-based transformer predictor $\mathbf{T}$ is used to recover the full indices map $\hat{d}$ from the received masked $d'$. Inspired by MAGE [31], we employ the encoder-to-decoder transformer structure for this predictor. In addition, using the continuous-domain MLIC decoding process, a continuous-domain latent $\hat{y}_c$ is recovered, which is fed into the mask predictor to guide the generation of the missing tokens. The idea is analogous

to the audio-to-text translation [27, 29, 39] where audio information is used to guide the text token generation through cross-attention. We insert a cross-attention module in each decoder block of the transformer decoder into which the continuous-domain latent feature is fed as guidance to assist discrete token generation.

**Pixel Decoder.** To merge the dual data streams for high-quality reconstruction, a duplicate pixel decoder is introduced alongside the VQ-Decoder $\mathbf{D_{VQ}}$. From the recovered $\hat{d}$, the vector-quantized latent $\hat{y}_{VQ}$ can be retrieved from the codebook $\mathbf{C}$ comprising the corresponding codewords indicated by $\hat{d}$. In previous efforts [21, 35], the VQ-Decoder reconstructs output $\hat{x}$ based on $\hat{y}_{VQ}$ alone. In our approach, the duplicate decoder serves as a correction network, which takes as input the recovered continuous-domain latent $\hat{y}_c$, and sends the decoded representation from each up-sampling layer of the duplicate pixel decoder to the corresponding up-sampling layer of the VQ-Decoder to rectify the biases. Such rectification imposes important fidelity information from the continuous domain to the reconstructed image, providing high perceptual quality and high fidelity simultaneously.

## 3.2 Complexity-aware Dynamic Masking

Image regions have uneven detail complexity. By adjusting bits allocation to different regions according to their complexity can further improve the compression efficiency. We analyze the complexity score of image regions using several metrics, including entropy, contrast, color diversity (histogram entropy), and spatial frequency (in Fourier domain), similar to ClassSR and its further extensions [25, 34, 45]. Each metric is normalized to be in range [0,1] over the training data, and the normalized metrics are averaged together to give a final complexity score. Then for each image region, a dynamic mask schedule is set based on the image complexity score. A low mask ratio is assigned to regions with rich complex details to preserve the intricate information, while a high mask ratio can be used for simple regions for extreme bit reduction.

## 3.3 Training Pipeline

To effectively train the proposed HybridFlow framework, our training process is divided into three stages, as described in Figure 2.

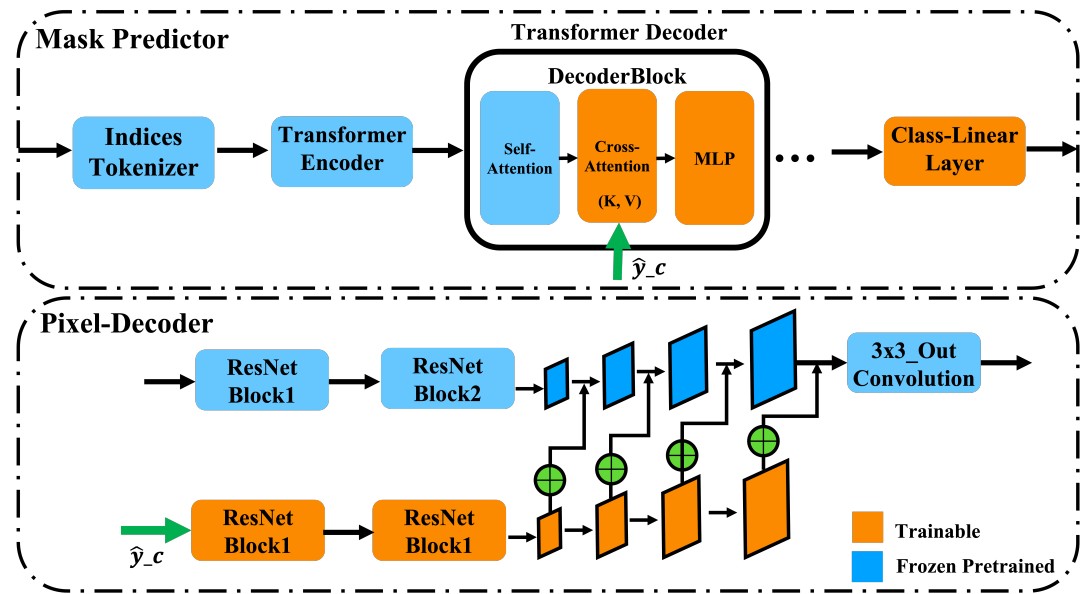

Figure 2: Training pipeline for our proposed framework.



**2_4**  **1_4**  **1_9**  **1_16**

🟥 **Remaining Indices**  ⬛ **Masked Indices**

Figure 3: Candidate mask schedules for codeword indices map (shown in the 8x8 indices map). The average mask ratio for each schedule is 50%, 75%, 90.3%, 93.75%, respectively.

**Pre-training.** The continuous-feature based LIC encoder to generate the continuous latent feature and hyperpriors used the pre-trained MLIC encoder [23]. The codebook-based VQ-Encoder and the learned visual codebook to generate the codebook-based latent feature, as well as the VQ-Decoder, used the pre-trained VQGAN model [11]. These pre-trained modules aimed to compute the dual-stream latent features that emphasized on high-quality reconstruction and high-fidelity reconstruction respectively.

**Transformer Predictor Training.** The pre-trained modules in the previous stage were kept frozen, and we trained the transformer predictor in this stage. For each decoder block in the transformer predictor, we selectively trained our cross-attention module, the MLP module, and the outermost MLP structure responsible for mapping to token logits, while maintaining the pre-trained parameters related to self-attention unchanged. We followed the training design of MAGE, where we randomly masked parts of the token sequence. Let $Y$ denote the flattened token output (indices) from the VQGAN Codebook, $M_b$ denote a randomly generated binary mask, $Y_m$ denote all masked tokens and $Y_r$ denote the remaining unmasked tokens. The transformer predictor were trained to accurately predict the masked tokens. The loss function between

the probability distribution of predicted masked tokens and the corresponding ground truth was formalized as:

$$L_{prediction} = -E(\sum \log p(m_i|Y_r)), \tag{1}$$

where $m_i \in Y_m$ is the predicted masked tokens. Following the previous MIM works [7, 15, 31], we calculated the loss only on the masked proportion for better model capacity.

**Pixel Decoder Training.** The modules mentioned in previous stages were kept frozen, and we trained the pixel decoder in this stage. We first replicated an identical pixel decoder from the pre-trained VQGAN decoder [11] as our duplicate decoder, which served as a correction network. Then we finetuned the duplicate decoder by minimizing the pixel-level loss between the original input $x$ and the reconstruction $\hat{x}$:

$$L_{pixel\_distortion} = w_1 * L1(x, \hat{x}) + w_2 * L_{perceptual}(x, \hat{x}), \tag{2}$$

$w_i$ being the loss weights, $L1$ being the L1_loss and $L_{perceptual}$ being the perceptual loss generated by AlexNet. The VQ-Decoder from the pre-trained VQGAN model [11] were kept frozen so that the pixel-level loss enabled the duplicate decoder to refine pixel fidelity without excessively compromising the perceptual quality obtained by the codebook-based representation.

## 4 EXPERIMENT RESULTS

### 4.1 Experiment Settings

**Datasets.** We trained our method using ImageNet, with over one million diverse images to fully utilize the modeling capability of our dual-stream system. For performance evaluation and fair comparison with several existing methods [5, 6, 9, 23, 35], we conducted tests on the Kodak [1], CLIC 2020 test set [12] and Tecnick [2] dataset that were used by the previous methods.

**Model Configurations.** As described in Section 3.3, for continuous-feature-based data stream, the pre-trained MLIC model with the lowest quality ($\lambda = 0.0018$ as described in [19]) was used. For the codebook-based data stream, we used the pre-trained VQGAN [11] with transformer mask predictor from MAGE [30]. The parameters of the proposed bridging modules in decoder were trained according to the training strategy of Section 3.3.

In detail, to match the default 256-length token indices input of the pre-trained MAGE model, we fed the 256x256x3 image patches into our system. This ensured that the length of the flattened indices map was 256. The additional complexity-aware module used three complexity score thresholds to select from three mask schedules: the 1_9 mask schedule for easy regions with complexity score < 0.24, the 1_2 mask schedule for complicated regions with complexity score > 0.77, and the 1_4 mask schedule for medium regions in between. These thresholds were empirically determined based on the training data as described in Section 3.2.

**Mask Prediction Inference.** Unlike prior token-based transformers using random sampling to create novel content, our decoding relied on max logits to remove the randomness for stable generation to serve the compression purpose. Additionally, different from iterative decoding used by MAGE [31] and MaskGIT [7], our inserted cross-attention module allowed accurate predictions in a single forward pass during testing, eliminating the need for multiple step-by-step recoveries. These changes reduced randomness in mask token prediction and improved decoding efficiency.

## 4.2 Effectiveness of Bridging Mechanism

We first validated the effectiveness of the proposed bridging mechanism in two aspects: Using the continuous-domain hyperprior information to assist the transformer predictor in recovering the original indices map, and using the fidelity information from the continuous latent feature to help the pixel decoding through the duplicate decoder.

**Continuity-assisted Mask Predictor.** We compared our proposed transformer predictor and the original pre-trained MAGE on the ability to recover the original indices map based on just a portion of the ground-truth indices. Several mask schedules were tested, and the recovered indices map was directly fed into the pre-trained VQGAN decoder for image restoration. As shown in Figure 4, compared to the ground truth, the predictions of the original MAGE exhibited increasing deviation as the mask ratio grows. In contrast, our transformer predictor achieved highly stable predictions that remained faithful to the features of the original image, thanks to the global visual cues from the continuous stream introduced by the cross-attention module.

**Continuity-assisted Pixel Decoder.** We compared the proposed continuity-fused pixel decoder and the original pre-trained VQGAN decoder. The indices map was unmasked in this test and was directly fed into the pixel decoders to solely evaluate the effectiveness of the proposed pixel decoder. As shown in Figure 5, decoding pixels based on codebook-based latent feature alone might not align well with the general pixel distribution, especially in sensitive areas like faces or texts, due to the limited amount of codeword features. The continuous features carried important fidelity information about the original distribution, which provided to the pixel

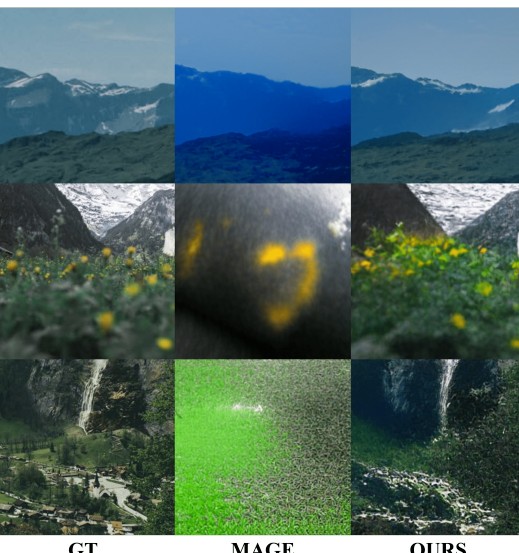

**Figure 4: Effectiveness of our mask predictor. The first to last row separately have a mask schedule of 1_4, 1_9, 1_16.**

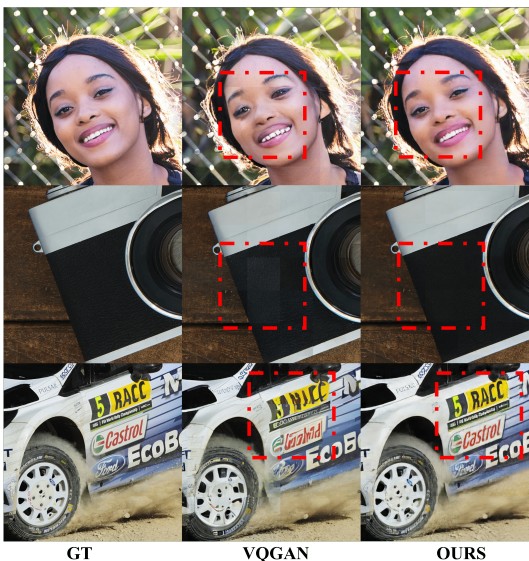

**Figure 5: Effectiveness of continuity-assisted pixel decoder. The red boxes emphasize specific regions where the duplicate decoder leverages continuous-domain information to effectively correct codebook-based deviation.**

decoder fidelity-preserving rectifications in difference-sensitive regions. As a result, the generated images could better match the original image content, while maintaining high perceptual quality at the same time.

## 4.3 Effectivness of Complexity-aware Masking

As mentioned in Section 3.2, different mask schedules were applied to modules with varying complexities. That is, while maintaining a similar image quality, more transmission bits were allocated to high-complexity (often perception-sensitive) regions, and fewer bits were used for low-complexity areas. As shown in Figure 6,

based on the complexity of image patches obtained through partitioning, a large number of blocks with simple features were defined as "easy", while regions sensitive to features, such as faces and intricate clothing patterns, were categorized as "tough" ("complicated"). The complexity-aware module offered a straightforward yet effective way to further reduce transmission bits, *e.g.*, with an additional 12.5% bpp reduction on average compared to the uniform 1_4 masking schedule, enabling our framework to better serve the extreme low-bitrate image compression scenario.

## 4.4 General Performance Comparison

**Compared methods.** To demonstrate the advantages of our proposed dual-stream HybirdFlow image compression framework in extremely low-bitrate scenarios, we compared our work with single-stream VQGAN [11] and single-stream MLIC [23] (as SOTA continuous-feature-based LIC). This comparison effectively showed the balanced overall performance improvement obtained by the fusion of the dual streams. Additionally, we compared our method with finetuned VQGAN compression [35], another codebook-based LIC approach, and the SOTA traditional compression method VVC [6].

**Evaluation Metrics.** In terms of evaluating image compression models with low bitrates, prior works often either measure pixel-level differences (PSNR) or measure perceptual differences (LPIPS), while seldom both. Actually PSNR accentuates visual resemblance when viewed by human eyes, while LPIPS tends to emphasize local image quality, including clarity and specific details. Both metrics are practically crucial for image compression tasks.

In this experiment, we showed that even in the context of extreme low-bitrate image compression, our proposed dual-stream structure could achieve a good balance between both metrics. We could maintain image clarity and expressive details without significant deviation from the original image.

**Qualitative Comparison.** In this experiment, our mask module adopted a fixed 1_4 mask schedule to maintain a consistent relationship between model compression quality and bitrate for fair comparison. As illustrated in Figure 8, our approach exhibited significant advantages in image reconstruction quality over single-stream MLIC and VQGAN in extremely low-bitrate scenarios. The reconstruction results of MLIC presented notable oil-painting-like blurs with regular occurrences of abnormal noise. Although reconstructions based on VQGAN were clear overall, they showed significant pixel deviations in detail-sensitive regions, especially around edges of image patches, leading to noticeable pixel discontinuities that severely affected visual perception. Moreover, for images containing textual information, the limited features of VQGAN caused local distortions in text content. In contrast, our approach visually balanced clarity and fidelity by ensuring image reconstruction clarity while largely reducing blur and noise, and by correcting pixel distortions in VQGAN through the complementary structural information from the continuous stream.

**Quantitative Comparison.** Our method provides different compression rates ranging from approximately 0.025 to 0.065 bpp through a variable mask schedule. The lowest quality corresponds to masking out all codebook indices, and the highest quality corresponds to no mask at all. As shown in Figure 7, traditional methods such as VVC [6], as well as single-stream continuous-feature-based

LIC like Cheng2020 [9] and MLIC [23], usually performed better than single-stream codebook-based LIC methods like [11] in terms of pixel-level PSNR. However, their performance in terms of perceptual quality like LPIPS was poor. They tended to prioritize the overall pixel similarity by generating large, detail-less, blurry patches, thereby sacrificing image clarity and fine details. In comparison, the single-stream codebook-based LIC methods had poor PSNR performance in general (about 4.5 dB lower than traditional methods at the same bitrate). This confirmed that reconstruction solely based on learned codebook introduced significant pixel-level distortions. However, due to the pixel decoder's reliance on high-quality codebook features, the reconstruction was quite stable, resulting in much better LPIPS. Our method combined the advantages of both approaches, and achieved a good balance between perceptual LPIPS and pixel-level PSNR. Compared to single-stream codebook-based LIC, our PSNR curve closely resembled traditional methods, enhancing the average PSNR by 3.5 dB with even further improvement on LPIPS. Compared to traditional methods that solely focused on PSNR performance, although our PSNR is lower, the reconstructed image had significantly better perceptual quality and clarity with an average LPIPS increase of 55.7% across three testsets . These results demonstrated that for extreme low-bitrate image compression scenarios, our HybridFlow approach provided a good balanced image reconstruction quality for practical applications.

**Boundary Effects on Image Partitioning.** To reduce memory and computation requirements of the image compression pipeline, it is a common practice to partition large images into smaller blocks, *e.g.*, $256 \times 256$ for efficient individual processing. This strategy usually causes discrepancy in pixel continuity, leading to discrepancy in latent feature continuity that results in explicit block boundaries in reconstructed images. Traditionally, additional post-processing module such as a smoothing network is needed to alleviate this problem. However, with extreme low bitrates, solely codebook-based compression methods like VQGAN [11] presents severe boundary effects due to dramatic quantization effects in the visual space using the limited number of codewords, which is difficult to be trivially solved by post-processing (as shown in the zoomed-in image patches in Figure 8). In contrast, by using continuous latent features to correct the pixel decoding process, our method adaptively adjusts for saturation biases and rectifies structural latent features within the image, and substantially reduces the boundary effects without using any explicit post-processing modules.

## 4.5 Conclusion

In this paper, we propose a dual-stream HybridFlow framework, tailored for ultra-low-bitrate image compression. By integrating continuous-domain features into the discrete-domain representation, we provide high perceptual quality and high fidelity in reconstructed images simultaneously with extreme low bitrates. We also selectively mask the indices map to further reduce information rates. Specifically, we introduce a token-based transformer with cross-attention modules to incorporate guidance from the continuous domain efficiently, enabling us to predict full indices maps based on partial masked indices and maintain fidelity to the original distribution. We also infuse continuous-domain features into the

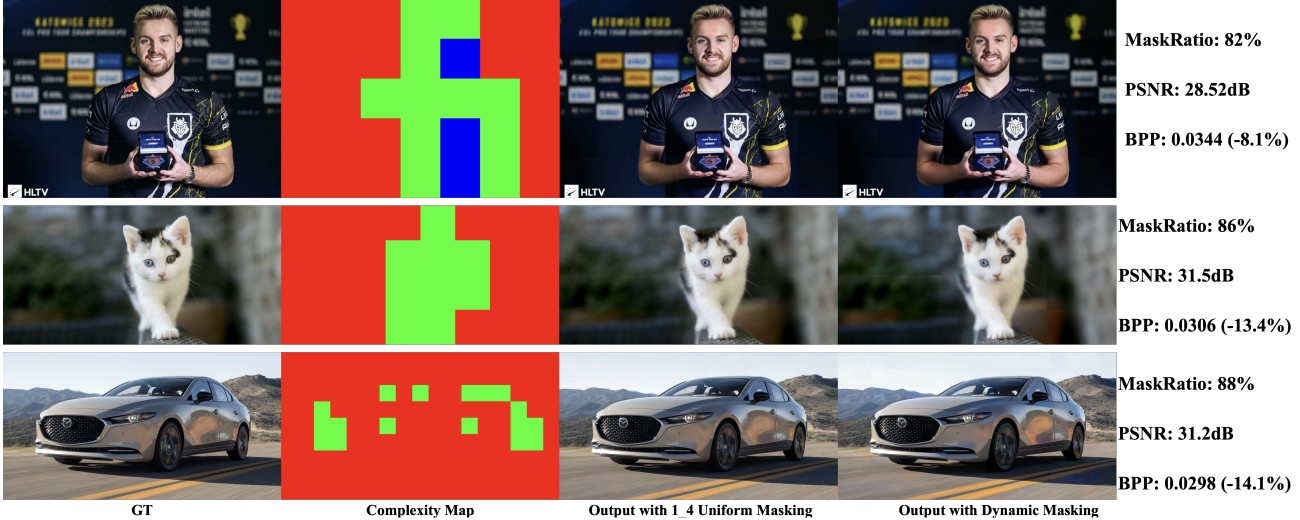

MaskRatio: 82%

PSNR: 28.52dB

BPP: 0.0344 (-8.1%)

MaskRatio: 86%

PSNR: 31.5dB

BPP: 0.0306 (-13.4%)

MaskRatio: 88%

PSNR: 31.2dB

BPP: 0.0298 (-14.1%)

| GT | Complexity Map | Output with 1_4 Uniform Masking | Output with Dynamic Masking |

**Figure 6: Effectiveness of the complexity-aware dynamic masking scheduling. The red, green, and blue color represent "easy", "medium", and "tough", respectively. The rightmost column depicts the image quality in PSNR of the outputs with dynamic masking and the percentage of reduction in bpp compared to 1_4 uniform masking.**

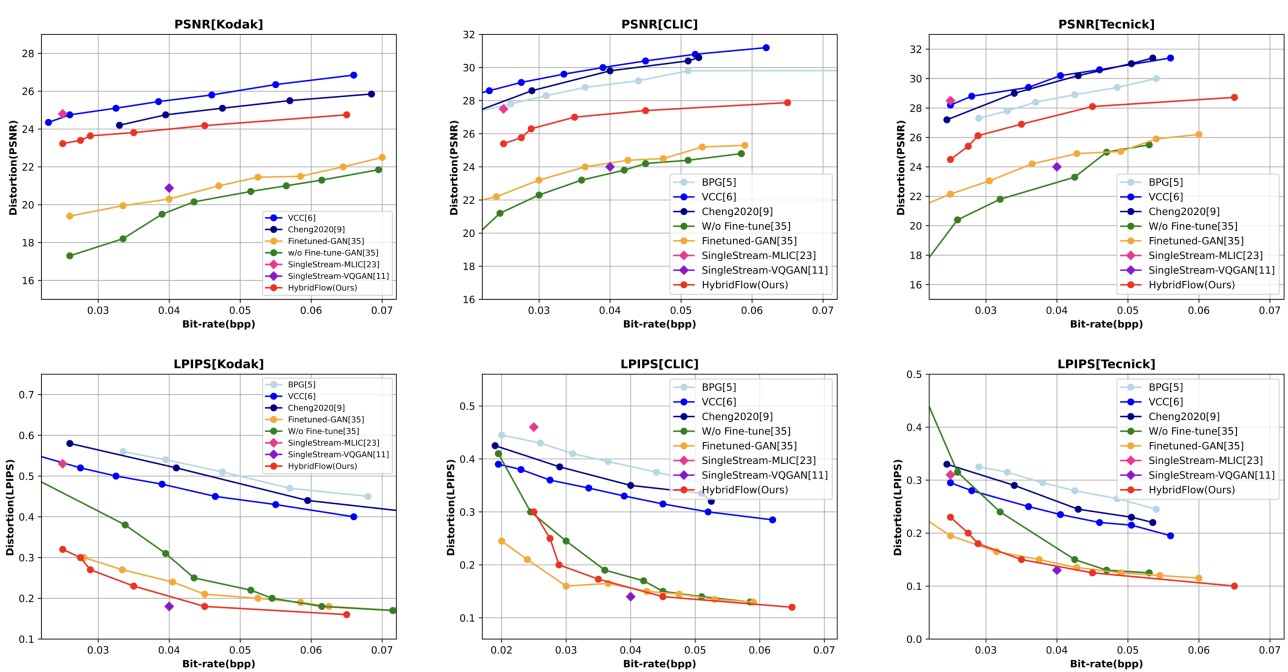

**Figure 7: Quantitative results on Kodak, CLIC2020, Tecnick datasets. (PSNR the higher the better, LPIPS the lower the better)**

pixel decoder through a correction network, which reduces pixel-level distortions in reconstruction, achieving both high perceptual quality and high fidelity. Finally, we offer an optional complexity-aware module to select different mask schedules for different image patches, allocating limited bits more efficiently. Experimental results demonstrate the robustness of our method across various datasets, with significantly improved PSNR and similar or even better LPIPS compared to existing codebook-based LIC methods, and with significantly improved LPIPS compared to continuous-feature-based LIC methods. Our approach provides a general dual-stream LIC framework for building bridges between the continuous and the codebook-based feature domains, further advancing research in ultra-low-bitrate image compression.

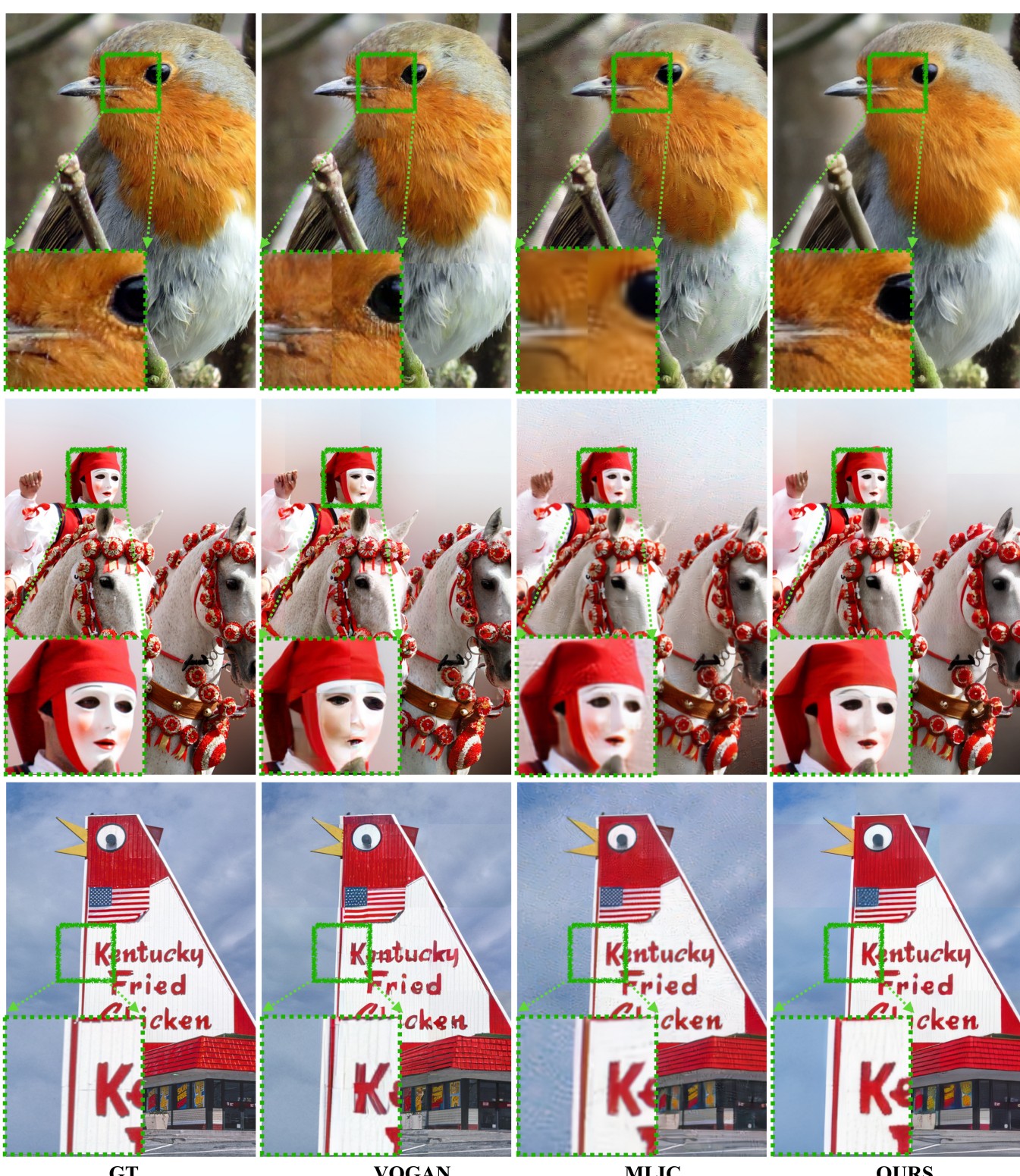

GT        VQGAN        MLIC        OURS

**Figure 8: Qualitative visualization of single-stream-based VQGAN, MLIC and our proposed dual-stream results. (Zoom-in for better visual comparison)**

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
