# OpenReview forum: "HybridFlow: Infusing Continuity into Masked Codebook for Extreme Low-Bitrate Image Compression"
_acmmm.org/ACMMM/2024/Conference — MM2024 Poster_

### Official Review · Reviewer_k5cu · 2024-05-23

**Rating:** 3
**Confidence:** 3

**Summary:**

The authors propose a hybrid framework that benefits from the dual-stream complementarity of the two categories of technologies: the continuous-feature-based and codebook-based streams. The codebook-based stream benefits from the high-quality learned to simultaneously achieve extremely low bitrate transmission and high-quality reconstruction. Two parallel streams are generated from the input image: one is a high-quality codebook-based discrete index that exploits learned general image priors to achieve high perceptual reconstruction quality; the other is extremely low bitrate continuous features that provide fidelity of detail flow. The two streams are combined through an efficient bridging mechanism for mask token generation and corrective pixel decoding

**Strengths:**

1. This paper proposes a novel dual-stream framework, HyrbidFlow, which combines the continuous-feature-based and codebook-based streams.
2. The author introduces a masked token-based transformer to only transmit a masked portion of codeword indices, which can save bits.

**Limitations:**

1. The evaluation of the codecs in this paper excludes two important metrics:  the coding time and complexity. The transformer integrated into the model will significantly increase the complexity of the model, causing it hard-trained and deployed.
2. In terms of qualification evaluation, the proposed codec has no advance in psnr compared to VVC and cheng2022, and the performance on LPIPS is not yet significant.  The ultra-low bitrate has also been clearly shown according to Figure 7, where all codecs can decode images with the lowest bpp (0.02- 0.03). Even in the near 0.02 bpp, other codecs perform better and show lower LPIPS.
3. Many important baselines have not been compared. Besides other superb codecs like JPEGXL,  EILC, and flow-based have not been compared in this paper. So the performance of this method has not been powerly proven.

**Suitability:**

3

---

### Official Review · Reviewer_LSWN · 2024-05-24

**Rating:** 4
**Confidence:** 4

**Summary:**

A method for ultra-low bit rate compression that combines continuous and discrete features to reconstruct high-quality images. This method uses continuous features to assist the Transformer in predicting discrete features while helping to correct pixel level errors during the Decoder reconstruction

**Strengths:**

• As demonstrated in the supporting materials, continuous features can significantly improve the prediction accuracy of discrete features.
• This method innovatively proposes using continuous features to provide a prior distribution for the Transformer, assisting in predicting discrete features to reduce quantization errors
• Continuous features are used to assist the decoder in correcting pixel level errors, compensating for the disadvantage of the decoder's inability to ensure content consistency in ultra-low bit rates.

**Limitations:**

Although this is an innovative approach, there are still several shortcomings:
• First, the paper utilizes a majority of pre-trained models but lacks comparisons with these models. For example, the proposed method is only compared with the VQGAN model using a single codebook.
• Second, the ablation experiments are insufficient. When validating the effectiveness of the bridging mechanism, there are only a few experimental result images, lacking specific metric data and further experiments. For instance, presenting the contributions of continuous features to the prediction of discrete features in the form of heatmaps would be beneficial.
• Third, the supplementary materials suggest that although the prediction accuracy remains at a relatively low level, high-quality reconstruction is still maintained. This is a noteworthy point that merits further discussion in the paper.

Minor Points:
1.	Several typos such as Figure7. VVC not VCC

**Suitability:**

2

---

### Official Review · Reviewer_cTPe · 2024-06-03

**Rating:** 4
**Confidence:** 4

**Summary:**

The authors propose an image compression codec targeted specifically for low bitrates. They introduce a novel approach by considering information from two separate flows: vector quantized and integer quantized (“continuous”), which takes advantage of the high semantic quality of vector quantization-based methods while retaining pixel-level fidelity due to the “continuous” information flow. The two flows are merged via a novel bridging mechanism which injects the “continuous” information into the vector-quantized features when decoding to pixel space. The authors also introduce a masking mechanism and corresponding prediction module to further reduce the bitrate, expanding the method’s applicability in the low-bitrate scenario. Here they use masked image modeling to predict codebook indices that have been masked according to a curated masking strategy, or optionally complexity-dependent based on the content of the image. Here the “continuous” features are also used to assist in the prediction of masked codebook indices. The authors validate the effectiveness of their approach with comprehensive ablations and comparisons on multiple datasets to other existing methods and show a quantitative and qualitative improvement in image quality.

**Strengths:**

* The hybrid information is interesting and novel, regarding both the dual-stream approach to use integer- and vector-quantized information and the proposed bridging mechanism to combine the two flows.
* Masking of codebook indices is a nice way to control bitrate without retraining the model, and conditioning the mask predictor on the integer quantized information seems to work well
* The authors perform extensive justifications and ablations to validate the effectiveness of their method.

**Limitations:**

* Comparisons should be performed with MS-ILLM, which performs very well in low-bitrate compression, or a discussion if a direct comparison is difficult.
* I feel that the methodology of combining the “continuous” and vector quantized streams in the decoder could be better explained. There is some explanation in the supplementary materials, but as this is a main contribution of the paper this should be discussed in the main text.
* Additional visual comparisons between the baseline would be appreciated.
* (Minor point), but perhaps “continuous” features is not the best terminology, as all features are discretized for entropy coding. I understand after reading that the difference is between vector quantized and integer quantized representations but using “continuous” can be misleading.

**Suitability:**

2

---

### Official Review · Reviewer_64j3 · 2024-06-04

**Rating:** 4
**Confidence:** 3

**Summary:**

In order to improve the image coding efficiency under ultra-low bitrate conditions, this paper proposes a hybridflow network, which could improve the coding performance by combining LIC based on continuous features and LIC based on codebooks. The performance of the framework is evaluated on three popular test datasets. Experimental results indicate the proposed framework could improve the perceptual performance obviously, preserving the signal quality level (such as PSNR).

**Strengths:**

1. The combination of continuous features and codebook indices to improve the coding performance for ultra-low bitrate is interesting.
2. The proposed complexity-aware dynamic masking is also effective to further reduce the coding expense.
3. The perceptual quality improvement of the proposed framework is obvious compared with other methods, based on the showcases.

**Limitations:**

1. There are two main parts in the bitstream, the continuous feature and the codebook indices. How the proportions of the two parts are decided, under various compression ratios?
2. Following 1, if the proportions of the two parts are fixed, why? If not, what's the strategy of the proportions setting and how it is derived.
3. For performance visualization, for example, in Figure 5 and Figure 8, the bitrate of various methods should be clarified, except for the bitrates are the same.
4. The showcases of non-LIC codecs (such as VVC) should also be presented, for a better comparison and understanding, such as in Figure 8.
5. What is the computational complexity of the proposed framework, especially compared with the base models (continuous feature codec and the codebook index codec, respectively).
6. Based on the loss function in equation 1 and 2, it seems that the base LIC models are already pre-trained, since no bitrate information is considered. What the performance would be, if the whole framework is further fine-tuned? It might further improve the performance.

**Suitability:**

3

---

### Meta-Review · Area_Chair_HeFY · 2024-07-07

**Recommendation:** Accept (Poster)
**Confidence:** 4

**Metareview:**

The paper proposes a dual-stream network, named HybridFlow, combining what the authors call  "continuous-feature-based" and "codebook-based" streams to achieve extremely low-bitrates image compression.  The idea of combining both streams from both Learned Image Compression (LIC) methods based on "continuous" features and on "codebooks" has the potential to improve the performance for such low-bitrate settings.  Also, the idea of masking codebooks indices is an interesting contribution of the paper.  The experiments are well-designed, and the ablation studies justify the proposed method.   In particular, the reviewers raised concerns regarding the need for more comparison to important baselines (e.g., MS-ILLM or GAN-based methods in general).

Given the interesting proposal and well-presented paper, the recommendation is to accept the paper.   We also recommend the inclusion of the required baselines in the final version of the paper.